# Putting the Value Back in RL: Better Test-Time Scaling by Unifying LLM Reasoners With Verifiers

**Kusha Sareen**
Mila, McGill University

**Morgane M Moss**
Mila, Université de Montréal

**Alessandro Sordoni**
Microsoft Research, Mila

**Rishabh Agarwal** *
Mila, McGill University

**Arian Hosseini** *
Google DeepMind, Mila

## Abstract

Prevalent reinforcement learning (RL) methods for fine-tuning LLM reasoners, such as GRPO or Leave-one-out PPO, abandon the learned value function in favor of empirically estimated returns. This hinders test-time compute scaling that relies on using the value-function for verification. In this work, we propose $RL^V$ that augments any "value-free" RL method by jointly training the LLM as both a reasoner and a generative Yes/No verifier using RL-generated data, adding verification capabilities without significant overhead. Empirically, $RL^V$ boosts MATH accuracy by over 20% with parallel sampling and enables $8 - 32\times$ efficient test-time compute scaling compared to the base RL method. $RL^V$ also exhibits strong generalization capabilities for both easy-to-hard and out-of-domain tasks. Furthermore, $RL^V$ achieves $1.5 - 2\times$ higher performance when jointly scaling parallel and sequential test-time compute with a long reasoning R1 model.

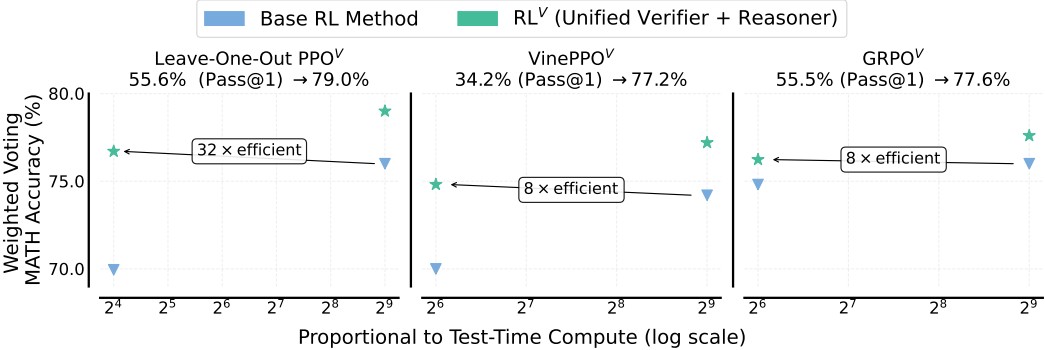

Figure 1: **$RL^V$ offers significant compute efficiency and performance gains** over base "value-free" RL methods when scaling test-time compute with weighted majority voting on MATH500 (Lightman et al., 2023). For scoring solutions, we use LLM-as-a-Judge as the verifier for the base method, while the trained unified verifier for $RL^V$. These results are based on RL fine-tuning Qwen2.5-Math-1.5B on Hendrycks MATH.

## 1 Introduction

Reinforcement learning (RL) on correctness rewards has emerged as a pivotal technique for advanced reasoning capabilities of large language models (LLMs) (DeepSeek-AI, 2025). A notable trend among state-of-the-art RL algorithms designed for LLMs, including GRPO (Shao

*Equally supervised. Corresponding authors: {kusha.sareen, agarwalr}@mila.quebec, arianhosseini@google.com

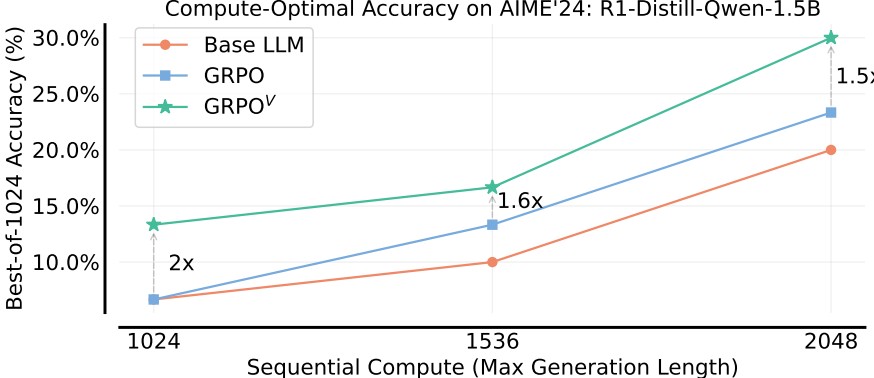

Figure 2: **Scaling Sequential and Parallel Compute Jointly** with GRPO$^V$ compared to baselines on the AIME'24 using R1-Distill-Qwen-1.5B as the base LLM. We use Hendrycks' MATH for RL fine-tuning. Each point represents the compute-optimal accuracy achieved at a sequence length using 1024 parallel samples. For GRPO$^V$, the compute-optimal strategy corresponds to Best-of-N selection guided by its unified verifier; for baselines, this strategy corresponds to majority voting or weighted voting using the same 1.5B model prompted as a verifier (LLM-as-a-Judge).

et al., 2024), VinePPO (Kazemnejad et al., 2024), and Leave-one-out PPO (Chen et al., 2025b; Ahmadian et al., 2024), is their shift from the canonical PPO (Schulman et al., 2017) method by abandoning the learned value function network, opting instead to rely on empirically estimated returns. This shift reduces both computational demands and GPU memory consumption, which is crucial for scaling RL training to increasingly massive LLMs.

Discarding the learned value function, while beneficial for RL training , sacrifices its potential utility at test time. Traditionally, the value function estimates expected future rewards, allowing it to serve as an *outcome verifier* (Cobbe et al., 2021) to assess correctness of a given reasoning chain. This verification capability is valuable for scaling inference compute through parallel search strategies like Best-of-N or weighted majority voting.

We argue that the potential for efficient test-time compute scaling offered by a value-like signal remains largely untapped in prevalent RL methods. To capture this potential without sacrificing training scalability, we propose RL$^V$ that augments "value-free" methods with a generative verifier (Zhang et al., 2024). Unlike traditional value functions predicting only scalar rewards, generative verifiers leverage the LLM's generation capabilities. Our core idea utilizes the abundant data generated during RL training to simultaneously train the LLM as a reasoner and a Yes/No verifier. Specifically, we jointly optimize standard RL objectives alongside a generative verification objective, framing verification as a next-token prediction task conditioned on the RL-generated reasoning sequences. This enables the *same* LLM to serve a dual function: acting as the policy generating solutions while simultaneously providing an intrinsic, generative score reflecting perceived solution correctness.

Empirically, RL$^V$ demonstrates significant advantages for test-time scaling. It boosts MATH accuracy by over 20% compared to the base RL method when using parallel sampling and enables substantially more efficient test-time compute scaling, achieving $8-32\times$ improvements, as shown with in Figure 1. Furthermore, RL$^V$ exhibits robust generalization capabilities, outperforming the base RL method not only on harder math problems in MATH[2] (Shah et al., 2025) but also on out-of-domain tasks like GPQA Physics (Rein et al., 2024), as illustrated in Figure 4. The benefits of RL$^V$ extend to long CoT reasoning models when scaling both parallel and sequential compute, where it achieves $1.5-2\times$ higher performance than baseline methods, consistently yielding the best results across different generation lengths and parallel sample counts (Figure 2, 6).

## 2 Related Work

**RL for reasoning.** Recently, there has been a research surge in eliciting improved reasoning from LLMs via RL, including traditional RL algorithms, such as PPO (Zeng et al., 2025). Notably, one can utilize the value model trained in PPO as a verifier for test-time search (Liu et al., 2023). However, the trend towards "value-free" RL (Shao et al., 2024; DeepSeek-AI, 2025; Kazemnejad et al., 2024; Ahmadian et al., 2024; Chen et al., 2025b) in recent LLM applications discards this possibility, and this method also involved the overhead of training a separate model. Our work aims to reintegrate verification with RL, proposing a simple approach where a generative verifier is trained concurrently with the policy, leveraging data generated during RL training.

**Test-time Verification.** Verification serves as a powerful approach for improving LLM reasoning by scaling test-time compute, often using separate models trained via binary classification (Cobbe et al., 2021; Luo et al., 2024; Yu et al., 2024; Lightman et al., 2023; Setlur et al., 2025; Zhang et al., 2025), preference learning (Hosseini et al., 2024; Yuan et al., 2024)), or more recently using next-token prediction (Zhang et al., 2024; Mahan et al., 2024; Ankner et al., 2024). However, these separate verifiers incur significant overheads: they demand large training datasets, extra compute cycles, and substantial GPU memory during inference, potentially limiting the size of the LLM reasoner when loaded together. In contrast, we jointly train a single LLM using RL and generative verification. Our method offers a capable verifier essentially for free, incurring no memory and very minimal compute cost. Furthermore, as seen in Figure 1, our approach results in much better inference compute scaling than using the RL policy as a verifier via LLM-as-a-Judge (Bai et al., 2022; Zheng et al., 2023; Chen et al., 2025a; Kim et al., 2024; Zhao et al., 2025)

## 3 Background

**Reinforcement Learning for LLMs** involves maximizing the expected reward under the LLM $\pi_\theta$ on a set of prompts $\mathcal{X}$, where we typically use a binary correctness reward (DeepSeek-AI, 2025). To maintain stability and prevent the fine-tuned LLM from deviating too much from the base LLM $\pi_{ref}$, a KL divergence penalty is often added with a coefficient $\beta$, yielding the objective $\mathcal{J}(\theta)$ (Stiennon et al., 2020):

$$\mathcal{J}(\theta) = \mathbb{E}_{\mathbf{x}\sim\mathcal{X}}[\mathcal{J}_{RL}(\theta;\mathbf{x})], \text{ where } \mathcal{J}_{RL}(\theta;\mathbf{x}) = \mathcal{J}(\theta;\mathbf{x}) - \beta D_{KL}[\pi_\theta||\pi_{ref}] \tag{1}$$

$\mathcal{J}_{RL}(\theta;\mathbf{x})$ is typically optimized using policy gradient methods, which we describe below.

**Proximal Policy Optimization (PPO)** (Schulman et al., 2017) is a canonical RL algorithm for fine-tuning LLMs, where gradient updates are constrained by a clipping mechanism to prevent large changes from the previous policy. The objective is described as:

$$\mathcal{J}_{PPO}(\theta;\mathbf{x}) := \mathbb{E}_{\mathbf{y}\sim\pi_{\theta_{old}}(\cdot|\mathbf{x})}\left[\frac{1}{|\mathbf{y}|}\sum_{t=1}^{|\mathbf{y}|}\min\left(p_t(\theta)\hat{A}_t,\ \text{clip}(p_t(\theta), 1-\epsilon, 1+\epsilon)\hat{A}_t\right)\right],$$
$$\text{where } p_t(\theta) = \frac{\pi_\theta(y_t|\mathbf{x}, \mathbf{y}_{<t})}{\pi_{\theta_{old}}(y_t|\mathbf{x}, \mathbf{y}_{<t})} \tag{2}$$

where $\epsilon$ is the clipping hyperparameter, and $\hat{A}_t$ is the advantage for token $t$. The advantage is typically estimated with GAE using a learned value network (Schulman et al., 2018). However, for LLMs this value network can be slow, memory intensive, and inaccurate, which has resulted in state-of-the-art methods discarding it.

**Group Relative Policy Optimization (GRPO)** (Shao et al., 2024) is a variant of PPO designed to mitigate some of its drawbacks, particularly for training LLMs. A key part of GRPO is that it foregoes the need for an explicit value model. Instead, it estimates the baseline for advantage calculation directly from the rewards of a group of $G$ outputs $\{\mathbf{y}_1, \mathbf{y}_2, \cdots, \mathbf{y}_G\}$

generated for the same prompt $\mathbf{x}$. The objective function is:

$$\mathcal{J}_{GRPO}(\theta; \mathbf{x}) := \mathbb{E}_{\{\mathbf{y}_i\}_{i=1}^G \sim \pi_{\theta_{old}}(.|\mathbf{x})} \left[ \frac{1}{G} \sum_{i=1}^G \frac{1}{|\mathbf{y}_i|} \sum_{t=1}^{|\mathbf{y}_i|} \min\left(p_t(\theta)\hat{A}_{i,t}, \text{clip}\left(p_t(\theta), 1-\epsilon, 1+\epsilon\right)\hat{A}_{i,t}\right) \right],$$

$$\text{where} \quad \hat{A}_{i,t} = \frac{r_i - \text{mean}(\{r_1, r_2, \cdots, r_G\})}{\text{std}(\{r_1, r_2, \cdots, r_G\})}, \ r_i = r(\mathbf{x}, \mathbf{y}_i)$$

**Leave-One-Out PPO** (Chen et al., 2025b) also drops the value network, similar to GRPO, and estimates the advantage using a leave-one-out estimator (Kool et al., 2019). Given $K$ outputs for a prompt, for each output, the advantage is estimated using the average reward of the remaining $K-1$ samples[1], that is, $\hat{A}_{i,t} = r_i - \frac{1}{K-1} \sum_{i \neq j} r_j$.

**VinePPO** (Kazemnejad et al., 2024) improves the credit assignment in PPO by computing unbiased Monte Carlo value estimates of intermediate states, instead of relying on an inaccurate value network. Specifically, it generates $K$ generations $\mathbf{y}'_k$ starting from state $s_t = \mathbf{x} \oplus \mathbf{y}_{<t}$ using the current policy $\pi_\theta$, and averages their reward to get the value estimate $\hat{V}_{MC}(s_t)$. This estimate is then plugged into the advantage calculation:

$$\hat{A}_t := r(\mathbf{x}, \mathbf{y}_{<t+1}) + \hat{V}_{MC}(s_{t+1}) - \hat{V}_{MC}(s_t), \text{ where } \hat{V}_{MC}(s_t) := \frac{1}{K} \sum_{k=1}^K r(\mathbf{x}, \mathbf{y}'_k) \quad (3)$$

**Test-Time Compute Scaling** A prominent research direction involves performing additional computation during test-time to boost the reasoning performance of LLMs. Prevalent techniques often scale compute in parallel by sampling multiple candidate solutions and employing heuristics like majority voting to select a final answer (Wang et al., 2023) or using a verifier for test-time reranking with Best-of-N (Cobbe et al., 2021) or weighted voting (Uesato et al., 2022). Recently, RL has enabled scaling compute sequentially by generating long chain-of-thought (CoT), characteristic of reasoning models like R1 (DeepSeek-AI, 2025).

**Generative Verifiers** Zhang et al. (2024) pose verification as next-token prediction, where the LLM takes a problem $\mathbf{x}$ and a candidate solution $\mathbf{y}$ as input and outputs a verification decision by predicting a token $c_\mathbf{y}$, which is either 'Yes' or 'No', to indicate correctness. Specifically, we train the verifier using a supervised fine-tuning (SFT) loss to maximize the likelihood of predicting 'Yes' for correct solutions $\mathbf{y}^+$ and 'No' for incorrect solutions $\mathbf{y}^-$:

$$\mathcal{J}_{Verify}(\theta; \mathbf{x}) := \mathbb{E}_{(\mathbf{x}, \mathbf{y}, \mathbf{I}, c_\mathbf{y}) \sim \mathcal{D}_{Verify}} \log \pi_\theta(c_\mathbf{y}|\mathbf{x}, \mathbf{y}, \mathbf{I}), \quad (4)$$

where $\mathcal{D}_{\text{Verify}} = \{(\mathbf{x}, \mathbf{y}^+, \mathbf{I}), \text{'Yes'}\} \bigcup \{(\mathbf{x}, \mathbf{y}^-, \mathbf{I}), \text{'No'}\}$ is a class-balanced verification dataset, and $\mathbf{I}$ corresponds to the prompt 'Is this solution correct? Answer Yes or No.'.

# 4 RL$^V$: Unifying Verifiers with "Value-Free" RL Reasoners

Prevalent value-free RL methods for LLMs (§3) improve training scalability but discard the value network, eliminating the intrinsic verification mechanism available in methods like PPO. This limits test-time compute scaling approaches that rely on a verifier to select the final solution among several candidates. Addressing this limitation currently involves suboptimal choices: deploying separate verifier models or value networks imposes significant overhead in data curation, compute, and GPU memory (Ahmadian et al., 2024), while prompting the base LLM as a verifier (LLM-as-a-Judge) has minimal overhead but is less effective due to its lack of task-specific training (Zhang et al., 2024).

Training LLM reasoners with RL already produce large quantities of solution data with correctness reward labels, used solely for improving the reasoning ability of LLMs. We propose to leverage this data for an additional purpose: using these solutions generated

---

[1]This coincides with many modern implementations of Reinforce Leave-One-Out (RLOO) (Ahmadian et al., 2024), including the one in the HuggingFace RLOOTrainer.

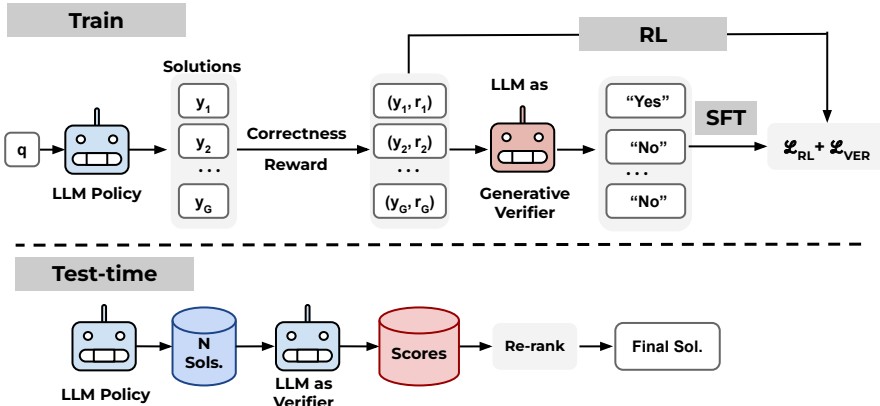

Figure 3: **Overview of RL$^V$**: (**Top**) During training, the LLM policy generates solutions $y$. This data is used for policy updates with RL and simultaneously trains the same LLM as a generative verifier via supervised fine-tuning (SFT) on correctness labels by asking the model 'Is this solution correct? Answer Yes or No'. (**Bottom**) At test time, the unified LLM generates N solutions and also acts as a verifier to assign scores for re-ranking using Best-of-N or weighted voting.

during RL to concurrently train a generative verifier within the same LLM used for reasoning. This approach, which we call RL$^V$, efficiently builds task-specific verification capabilities while avoiding the high memory and compute costs of separate verifiers and being more effective than LLM-as-a-Judge[2] based on prompting.

**Unified Training**. We train a single LLM, to perform both reasoning (problem-solving) and verification tasks. For each batch, the verification objective uses the (problem, solution, correctness reward) tuples generated during the RL process as training examples. Rather than employing separate prediction heads with regression or binary cross-entropy losses (alternatives explored in §5.4), we add the generative verification loss ($\mathcal{J}_{Verify}$ in Equation 4) to the RL fine-tuning objective ($\mathcal{J}_{RL}$ in Equation 1), resulting in this unified objective:

$$\mathcal{J}_{Unified}(\theta) := \mathcal{J}_{RL}(\theta; \mathbf{x}) + \lambda \mathcal{J}_{Verify}(\theta; \mathbf{x}), \qquad (5)$$

where the hyperparameter $\lambda$ balances the contribution of each objective. Specifically, the LLM learns to predict a 'Yes' or 'No' token to assess correctness of a given solution.

**Test-time Scaling**  At test time, we use the LLM verifier to score solutions generated by itself to guide the final answer selection. This score $s(\mathbf{x}, \mathbf{y})$ quantifies the verifier's confidence as its 'Yes' probability given the problem $\mathbf{x}$, solution $\mathbf{y}$, and prompt $\mathbf{I}$, that is, $s(\mathbf{x}, \mathbf{y}) := \pi_\theta(\text{Yes} \mid \mathbf{x}, \mathbf{y}, \mathbf{I})$. Here, we consider three parallel sampling approaches:

- **Majority Voting**: A verifier-free baseline that selects the most frequent answer.
- **Best-of-N**: Selects the solution with the highest verifier score $s(\mathbf{x}, \mathbf{y})$.
- **Weighted Voting**: Sum the verifier scores $s(\mathbf{x}, \mathbf{y})$ for solutions yielding the same final answer; select the answer with the highest cumulative score.

## 5   Experiments

We aim to investigate the effectiveness and characteristics of our proposed RL$^V$ method, which unifies a reasoner and a generative verifier within a single LLM. We answer several

---

[2]For Base RL ("value free") LLM-as-a-Judge experiments, we use the same verification prompt as RL$^V$ and only take the likelihood of predicting 'Yes' as the score in this setting as well, and do not generate verification CoTs for fair comparison.

key questions about this paradigm: 1) How does parallel test-time compute scale with $RL^V$? 2) How should the unified verifier be trained? 3) How should the unified verifier be used at test-time? 4) How does $RL^V$ interact with sequential scaling in thinking models?

**Setup** RL training for all our experiments utilized the Hendrycks' MATH dataset (Hendrycks et al., 2021) and were run on 4×A100 80G Nvidia GPUs. Evaluations are reported on MATH500 (Lightman et al., 2023), MATH$^2$ (Shah et al., 2025), GPQA (Rein et al., 2024), and AIME'24. For experiments involving shorter chain-of-thought (CoT) reasoning, we employed the Qwen2.5 Math 1.5B model (Yang et al., 2024). We finetune it with GRPO, Leave-One-Out PPO and VinePPO with and without unified verification. Training used a context window of 1024 tokens. During inference, we generated up to 1024 tokens for MATH500 and up to 2048 tokens for other test sets.

Long CoT experiments were conducted with DeepSeek-R1-Distill-Qwen-1.5B, a distilled version of DeepSeek-R1. We tune it with GRPO, and use SGLang (Zheng et al., 2024) for inference purposes. The RL training process involved sampling 32 problems per online iteration, generating 5 solutions per problem. With a batchsize of 8 this resulted in 32 updates per iteration. Training continued for 40 epochs. Models are trained using a "long" chain-of-thought prompt which encloses reasoning in special tags (see Appendix F.2).

## 5.1 Test-Time Compute Scaling With $RL^V$

**In-Distribution Generalization** $RL^V$ is up to 32× more efficient and achieves a 4% higher accuracy than baseline on MATH500 with 512 samples (Figure 4). Further, Figure 1 shows that across different RL methods, $RL^V$ achieves higher final accuracy levels. It also reaches strong accuracies with significantly less compute. Figure 8 shows these gains are maintained at the 7B scale.

**Easy-to-Hard Generalization** Zhang et al. (2024) and Sun et al. (2024) demonstrate that their trained verifiers can generalize to a fixed *separate* LLM reasoner on problem sets more challenging than those encountered during training. As illustrated in Figure 4 (center), our proposed method $RL^V$ with a unified reasoner and verifier, also exhibits strong easy-to-hard-generalization capacity on MATH$^2$ (Shah et al., 2025), containing much more difficult math problems that require a non-trivial combination of two distinct skills from MATH.

**Out-of-Domain Generalization** Going beyond easy-to-hard generalization, we also evaluate out-of-domain performance on GPQA Physics problems (Rein et al., 2024). $RL^V$ shows strong generalization resulting in more than 10% accuracy improvement compared to the baseline with 512 samples for weighted voting.

**$RL^V$ can lead to a better reasoner** Interestingly, we observe a sizeable positive transfer from unification to better pass@1 (Chen et al., 2021) performance (without any additional test-time compute) from unified training in $RL^V$ across all these tasks, suggesting a synergy between generative verification and RL objectives.

## 5.2 Compute-Optimal Scaling: How to Use Your $RL^V$ Verifier?

As outlined in §4, one can use a verifier at test-time to score generated solutions. Subsequently, a final answer can be selected using either the Best-of-N (BoN) strategy or Weighted Voting based on these scores. We conducted experiments on the AIME 2024 dataset with our GRPO$^V$ tuned models. Specifically, we tested variants based on Qwen2.5-Math-1.5B and R1-Distill-Qwen-1.5B, which generate short and long CoTs respectively (Figure 5).

We observe distinct behaviours depending on the model's CoT length characteristics. For the short CoT model (Qwen2.5-Math-1.5B, left panel of Figure 5), weighted voting consistently outperforms both majority voting and Best-of-N when sampling 8 or more solutions per problem. The trend differs significantly for the Long CoT model (Figure 5, right). While

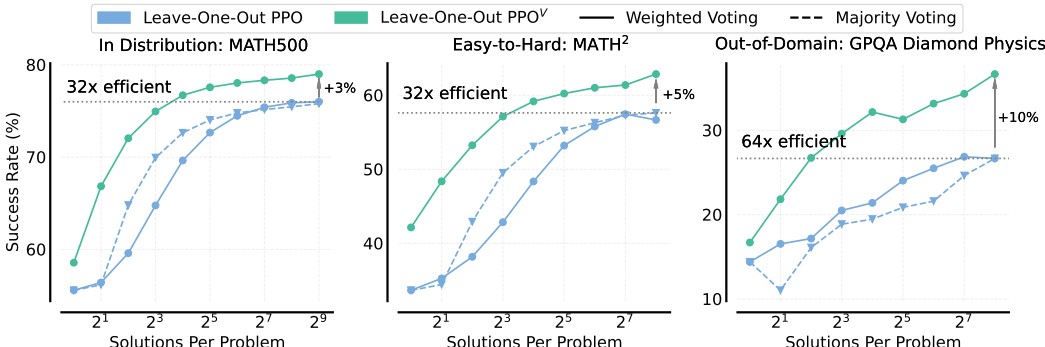

Figure 4: **RL$^V$ outperforms the base RL method** (Leave-One-Out-PPO) consistently across different number of solutions for different generalization settings with respect to the MATH training dataset. **(Left)** In-distribution Generalization on MATH500. **(Center)**. Easy-to-Hard Generalization on MATH$^2$ **(Right)**. Out-of-Domain Generalization on Physics problems in the GPQA Diamond split.

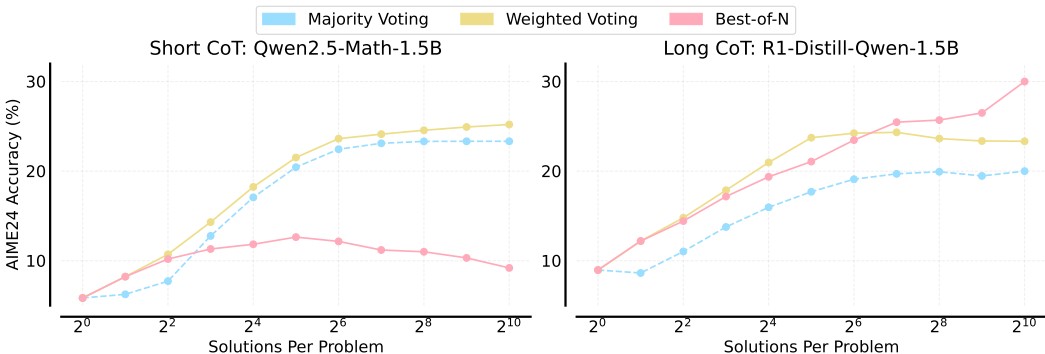

Figure 5: **Comparing Test-Time Compute Strategies** Evaluating verifier-based answer selection strategies (weighted voting, Best-of-N) and verifier-free majority voting on AIME'24. The optimal strategy differs for the short CoT and long CoT tuned models.

weighted majority still surpasses majority voting across all number of solutions, the Best-of-N strategy gets the highest accuracy for larger number of solutions ($\geq 2^7$) without showing signs of saturation. Our findings on the inference strategies are consistent with base RL experiments of the same models with LLM-as-a-Judge, where we do not train a unified verifier and simply prompt the same fine-tuned reasoning model as a verifier.

### 5.3 Complementing Long Reasoning Models with RL$^V$

An emerging technique to improve model performance at inference-time involves training them with RL to generate longer chain-of-thoughts (CoTs), simulating deeper reasoning before delivering an answer. Models employing this technique, such as DeepSeek-R1 (DeepSeek-AI, 2025) can dedicate extra sequential computation during inference to self-verify, reflect, and refine their final output. However, the amount of compute they allocate to find and verify a solution in their thought process is uncontrollable. Additionally, the exact mechanisms behind their reasoning are not yet fully understood.

Therefore, having a verifier which one can invoke reliably could further benefit inference time scaling. Our proposed method is complementary to sequential inference compute scaling. Figure 6 shows the AIME 24 success rate achieved by different methods when varying the number of solutions generated and the allowed generation length in tokens with budget forcing (Muennighoff et al., 2025). Notably, GRPO$^V$ method consistently achieves

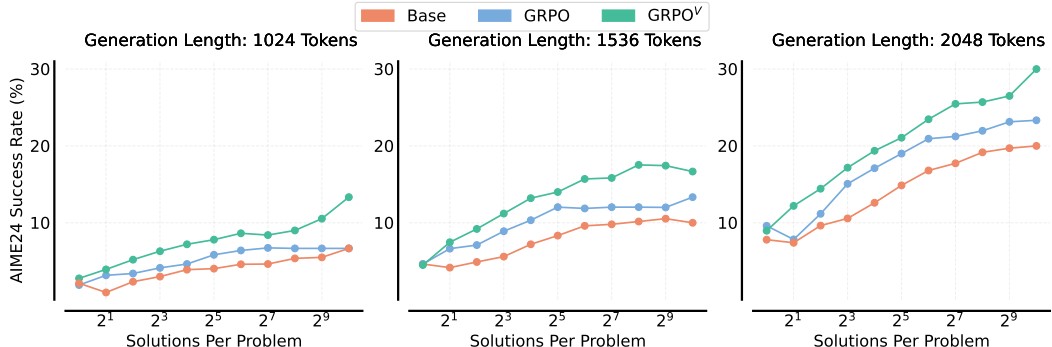

Figure 6: **Scaling Parallel Compute using RL$^V$ complements Sequential Scaling.** AIME'24 success rate *vs* number of solutions generated for Base (initial checkpoint), GRPO tuned model (no verification training) and GRPO$^V$ (unified verifier and reasoner) across varying generation lengths. GRPO$^V$ consistently outperforms GRPO, benefiting most from increased inference effort (more solutions, longer generation). For each model, we pick the inference-time strategy (Best-of-N, Majority Vote, Weighted Vote) that gives the highest success rate. See details of budget forcing in §D.

the highest success rate, more evident at longer generation lengths and scales well with number of sampled solutions, suggesting complementary gains to sequential scaling.

## 5.4 How To Train Your Unified Verifier?

The literature offers several options to train a unified verifier. Common approaches involve adding a dedicated verification head atop the policy network, trained using classification via binary cross-entropy (BCE) (Cobbe et al., 2021; Lightman et al., 2023) or regression objectives (Stiennon et al., 2020) to predict solution scores. Generative verification, proposed by Zhang et al. (2024), suggests it can produce capable verifiers without degrading, and sometimes even improving, the policy's core reasoning performance.

Figure 7 compares the Reasoner accuracy (measured by pass@1) and Verifier accuracy (measured on a balanced set of correct and incorrect solutions) of various verifier training approaches. Notably, Leave-One-Out PPO with separate verification heads performs poorly, both as a reasoner and verifier, compared to Leave-One-Out PPO$^V$. Overall, RL$^V$ outperforms base RL and RL methods with separate verification heads both as a reasoner and as a verifier. In addition, the RL$^V$ verifier accuracy is comparable to a separate verifier trained on logged data from the same RL run, showing minimal loss from joint training. While training a separate verifier on logged data can yield comparable or slightly higher accuracy, it incurs substantially greater training and inference costs.

A key hyperparameter in unified training is the verification coefficient $\lambda$ (see Equation 5), which balances the reasoning and verification objectives. Its impact is explored in Figure 7b for RL$^V$. For GRPO$^V$, a stark trade-off exists. Increasing $\lambda$ significantly boosts Verifier accuracy (from ~50% to ~80%), but drastically decreases Reasoner accuracy (from ~54% down to ~35%). Prioritizing verification heavily penalizes reasoning in this setup.

In contrast, for Leave-One-Out PPO$^V$, the trade-off is more nuanced. Verifier accuracy sees a consistent improvement until plateau as $\lambda$ increases (~50% to ~80%). Crucially, Reasoner accuracy peaks around $\lambda = 1$ before slightly declining. This suggests that Leave-One-Out PPO$^V$ can achieve a better balance, with optimal reasoning performance occurring at an intermediate verification coefficient where verifier accuracy is also strong.

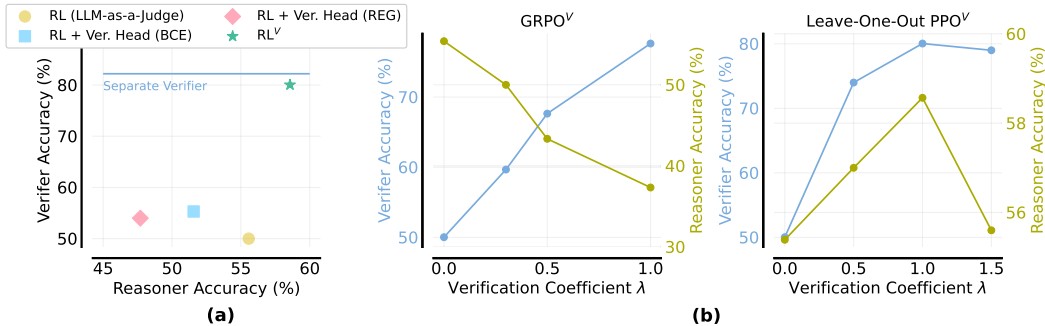

Figure 7: **(a) Comparison of Reasoner Accuracy versus Verifier Accuracy** for different unified verifier training strategies. Leave-One-Out PPO$^V$ significantly outperforms Leave-One-Out PPO using LLM-as-a-Judge and separate verification heads, with binary cross-entropy (BCE) or regression (REG) on both metrics. **(b) Impact of the Verification Coefficient** $\lambda$ on Reasoner and Verifier accuracy for RL$^V$. GRPO$^V$ shows a stark trade-off, while Leave-One-Out PPO$^V$ achieves a better balance with peak reasoner performance at an intermediate $\lambda$.

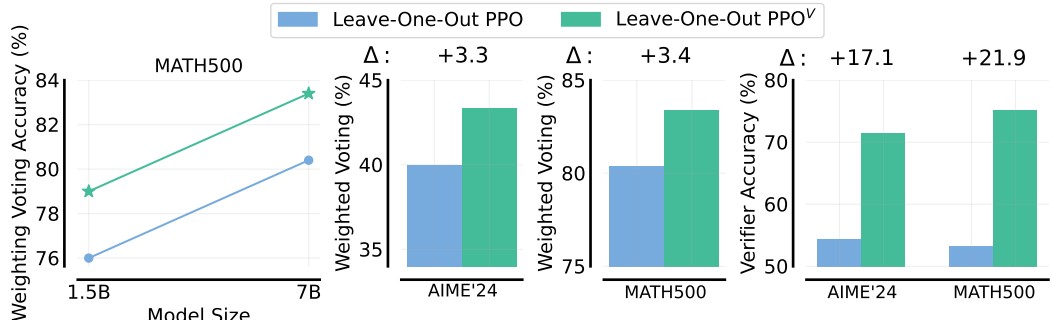

Figure 8: **RL$^V$ Scales with Model Size.** We observe consistent gains in weighted voting at 256 solutions for both 1.5B and 7B models. Additionally, training the 7B model for verification over simply using LLM-as-a-Judge results in drastically higher verifier accuracy on both MATH500 and AIME'24 benchmarks.

## 6   Discussion & Future Work

We proposed RL$^V$ that integrates verification into "value-free" RL frameworks without significant overhead, yielding substantial gains in reasoning accuracy, test-time compute efficiency, and generalization across MATH, MATH², GPQA, AIME 24 datasets. This method complements sequential scaling in long-reasoning CoT models and benefits from generative verification training, which proved superior to alternatives. The optimal test-time strategy (Best-of-N vs. Weighted Voting) can depend on model characteristics like CoT length, highlighting an interaction between the verifier and the reasoner's output style.

Building on these findings, future work could focus on enhancing the generative verifier to produce explicit CoT explanations (Zhang et al., 2024). However, training such a verifier necessitates verification-specific CoT data or RL training itself. We posit that future research could profitably investigate this direction to achieve a unified framework for both solution generation and verification through RL. Further investigation into RL$^V$'s applicability across other reasoning domains and scalability with larger LLMs, which was not possible due to our limited compute hardware, also remains pertinent.

## 7 Acknowledgements

We thank Khang Ngo for support in setting up infrastructure for experiments. We thank Siva Reddy and Arkil Patel for their valuable feedback on this paper and for engaging in insightful discussions. We also would like to express our gratitude to Mila's infrastructure team and Microsoft Research for providing the computing resources that made this project possible.

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

# Appendices

## A Test-Time Compute Strategies for Baseline

Similar to Figure 5, we evaluated base RL tuned (no verification training) variants based on Qwen2.5-Math-1.5B and R1-Distill-Qwen-1.5B, which generate short and long CoTs respectively, on AIME'24. The compute optimal strategy for GRPO$^V$ corresponds to Best-of-N (Figure 2). For base RL, Figure 9 shows that majority voting and weighted voting significantly outperform Best-of-N selection. We use the best performing inference strategy of GRPO in comparison to GRPO$^V$ in Figure 2 and Figure 6 to be fair.

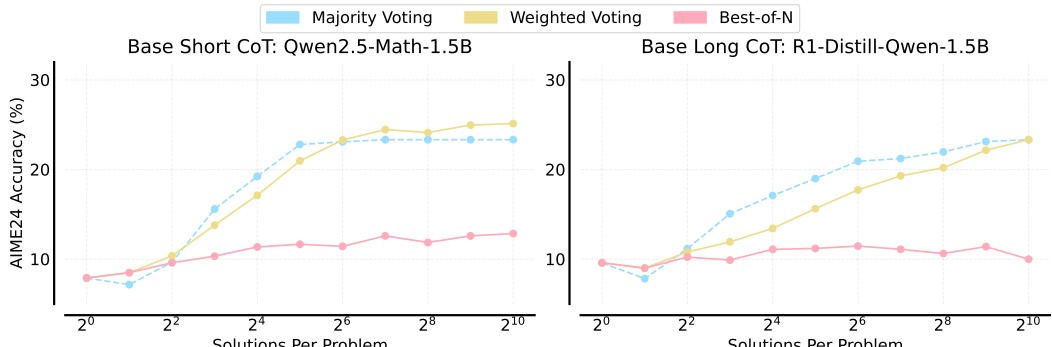

Figure 9: **Test-Time Compute Strategies for Base RL (No Verification Training) Methods** shows a different trend from Figure 5. LLM-as-a-Judge no longer provides high quality values, resulting in low Best-of-N scores. The compute optimal strategy here corresponds to majority voting and weighted voting.

## B Additional Experiments

**PPO Baseline** To compare with another RL method that learns a value function, we train Qwen2.5-Math-1.5B with PPO. Since PPO's value function provides token-level values and we need solution-level scores for re-ranking, we need to aggregate these values. We find averaging the values over the solution as opposed to taking the value of the last token gives the best results. Figure 10 shows results on MATH500. These results suggests that PPO's value function can act as a verifier but is less effective than RL$^V$.

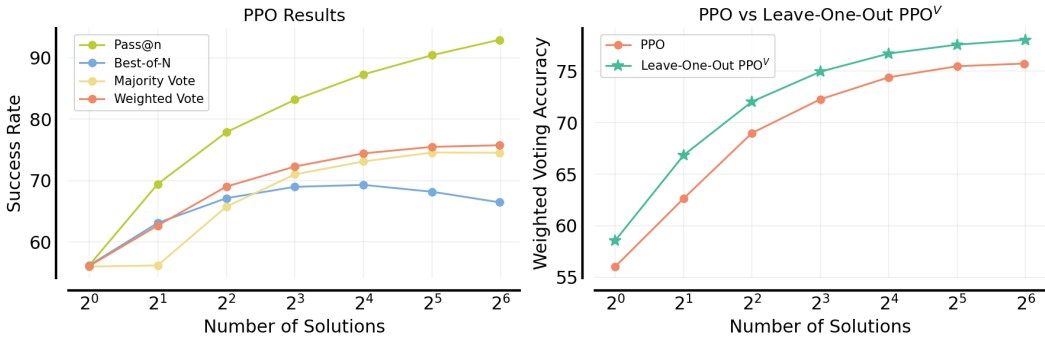

Figure 10: **PPO as a Verifier** shows the value function from PPO can act as a verifier, though it is less effective than RL$^V$.

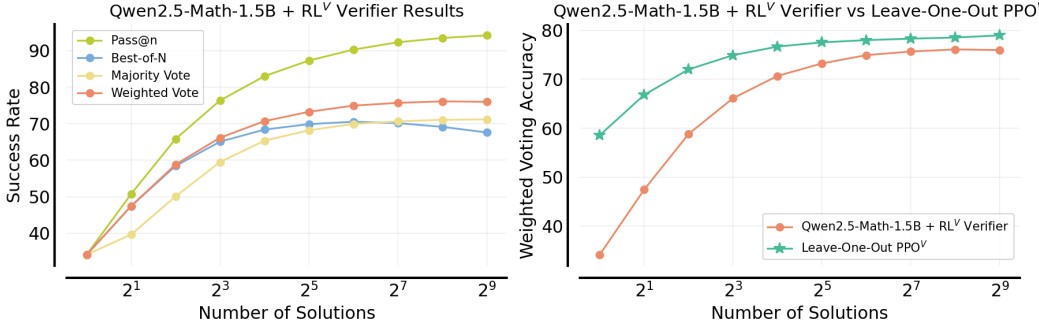

Figure 11: **RL$^V$ Verifier with Base Policy** shows that there is drastic improvement in reasoner and verifier quality during training and that our verifier generalizes well to the base model.

**RL$^V$ Verifier with Base Policy**   We add an ablation using the the RL$^V$ verifier with the Qwen2.5-Math-1.5B reasoner and Leave-one-out PPO$^V$ verifier during inference. The results follow in Figure 11.

This demonstrates that during RL$^V$, there is a drastic improvement in the quality of the reasoner itself (Pass@1: 34.1% to 58.6% in right figure) in addition to verification ability (weighted vote for Qwen2.5-Math-1.5B + RL$^V$ is 5% higher than majority vote in left figure). It also shows that Pass@1 is not the most important factor when scaling inference, since a capable verifier is able to bring Qwen2.5-Math-1.5B close to Leave-one-out PPO$^V$ at high inference compute budgets despite a 25% lower Pass@1.

This ablation also shows the generalization capabilities of our verifier since it has been trained on on-policy samples from the RL trained version of the model (not the solution distribution from the base model), yet it still works well.

**Summary Table**   Below in Table 1 we summarize some results comparing RL$^V$ with RL at 8 solutions per problem.

## C   Training Details

As follows are some details that enabled co-training. A linear ramp up was for both learning rate and verification coefficient, such that the maximum value is reached ¾ of the way through training. We do a hyperparameter search for learning rate for all methods and use standard settings for other hyperparameters. To ensure both reasoner and verifier objectives are trained jointly, we include both gradients in a single batch using gradient accumulation. For a given reasoner batch, the verifier gradient is computed by class-balancing right/wrong answers and computing the loss, oversampling the class with less right/wrong answers. We ensure the batch size is large enough that the resampling is reasonable. This ensures 100% of the data generated during RL is used for verification. We use an asynchronous infrastructure with 1 generation GPU and 3 training GPUs. All training for RL$^V$ and baselines is wall-clock matched.

## D   Inference Details

**Budget forcing**   The budget forcing implementation method can be described as follows:

1. Define the total budget for tokens, denoted by $k$.

2. Determine a buffer number of tokens, denoted by $b$.

3. During inference, generate up to $k - b$ tokens.

| Setting | RL Method | Task | Model | Success Rate (%) |
|---------|-----------|------|-------|------------------|
| Base | Leave-One-Out PPO | MATH500 | Qwen2.5-1.5B | 69.9 |
| | Leave-One-Out PPO$^V$ | MATH500 | Qwen2.5-1.5B | 74.9 |
| | VinePPO | MATH500 | Qwen2.5-1.5B | 60.3 |
| | VinePPO$^V$ | MATH500 | Qwen2.5-1.5B | 64.7 |
| | GRPO | MATH500 | Qwen2.5-1.5B | 68.8 |
| | GRPO$^V$ | MATH500 | Qwen2.5-1.5B | 69.8 |
| | PPO | MATH500 | Qwen2.5-1.5B | 72.3 |
| OOD | Leave-One-Out PPO | AIME24 | Qwen2.5-1.5B | 13.8 |
| | Leave-One-Out PPO$^V$ | AIME24 | Qwen2.5-1.5B | 14.2 |
| | Leave-One-Out PPO | AMC23 | Qwen2.5-1.5B | 55.1 |
| | Leave-One-Out PPO$^V$ | AMC23 | Qwen2.5-1.5B | 56.0 |
| | Leave-One-Out PPO | GPQA Physics | Qwen2.5-1.5B | 20.5 |
| | Leave-One-Out PPO$^V$ | GPQA Physics | Qwen2.5-1.5B | 29.6 |
| | Leave-One-Out PPO | MATH$^2$ | Qwen2.5-1.5B | 49.5 |
| | Leave-One-Out PPO$^V$ | MATH$^2$ | Qwen2.5-1.5B | 57.1 |
| 7B | Leave-One-Out PPO | MATH500 | Qwen2.5-7B | 76.4 |
| | Leave-One-Out PPO$^V$ | MATH500 | Qwen2.5-7B | 79.5 |
| | Leave-One-Out PPO | AIME24 | Qwen2.5-7B | 39.9 |
| | Leave-One-Out PPO$^V$ | AIME24 | Qwen2.5-7B | 43.3 |
| Long CoT | GRPO (2K length) | AIME24 | R1-Distill-1.5B | 15.1 |
| | GRPO$^V$ (2K length) | AIME24 | R1-Distill-1.5B | 17.9 |

Table 1: Comparison of Reasoner Success Rates across different RL methods, tasks, and LLMs, evaluated using 8 sampled solutions per problem.

4. To maintain the in-distribution characteristics of the initial generation, which will serve as the prompt for subsequent generation, truncate the generated text to the last completed sentence. If no full stops (.) are present in the initial generation, retain the entire generated text.

5. Append the conclusion tokens:

    ````</think>\n\n````

    to the resulting truncated text.

6. Use this concatenated text as the prompt for continued generation, allowing a maximum of $k$ tokens.

7. The final response is the concatenation of the truncated initial generation, the conclusion tokens, and the continued generation.

Thus, let $G_0$ be the initial generation of up to $k - b$ tokens, and $T(G_0)$ represent the truncated version of $G_0$ to the last completed sentence, or $G_0$ itself if no full stops are present. Let $C$ represent the conclusion tokens:

$$C = \text{````<think>\n\n````}$$

$G_1$ is the continued generation, using $T(G_0) \oplus C$ as the prompt, where $\oplus$ denotes concatenation. The final response $R$ is given by:

$$R = T(G_0) \oplus C \oplus G_1$$

where the total number of tokens in $R$ is constrained by the initial budget $k$.

This method balances the appropriateness of longer generation with the preservation of in-distribution properties by managing the initial generation and subsequent continuation.

# E Evaluation and Metrics

**Reliable Estimation of Best-of-N** To estimate the average success rate, we perform $M$ independent trials. In each trial, we draw $k$ samples out of $N(N > k)$ and select the best one (highest score). Finally, we average the success rates across these selected samples over $M$ trials (Hosseini et al., 2024).

$$\text{Best-of-}k := \frac{1}{\binom{N}{k}} \sum_{i=0}^{N-k} \binom{N-i-1}{k-1} \alpha_i \tag{6}$$

where $[\alpha_0, \alpha_1, ..., \alpha_{N-1}]$ are the binary correctness scores (0 or 1) for the candidate solutions sorted in decreasing order of their verifier scores.

# F Prompts

## F.1 Generating Solutions to Math Problems

| Math prompt |
|---|
| 1    Compute: $1-2+3-4+5- \dots +99-100$. |

## F.2 Generating Solutions to Math Problems with Long Chains of Thought

| Math prompt with long chain of thought |
|---|
| 1    Compute: $1-2+3-4+5- \dots +99-100$.
2
3    Think about the reasoning process in the mind and then provides an answer.
4
5    The reasoning process is enclosed within \<think\> \</think\> tags, i.e., \<think\> reasoning process here \</think\>. Put your final answer within \\boxed{{}}. |

## F.3 Verification Prompt

| Verification prompt |
|---|
| 1    Problem:
2    Compute: $1-2+3-4+5- \dots +99-100$.
3
4    Solution:
5    $(1-2)+(3-4)+ \dots +(97-98)+(99-100) = 50(-1) = \boxed{-50}.$
6
7    Is this solution correct? Answer with Yes or No. |

