# OpenReview forum: "Putting the Value Back in RL: Better Test-Time Scaling by Unifying LLM Reasoners With Verifiers"
_colmweb.org/COLM/2025/Conference — COLM 2025_

### Official Review · Reviewer_f9F3 · 2025-05-11

**Rating:** 6
**Confidence:** 3
**Ethics Flag:** 1

**Summary:**

This paper proposes RL-V, a method to enhance reasoning performance and efficiency for LLMs by reintegrating verification capabilities into “value-free” RL methods. Specifically, RL-V trains an LLM as both a reasoner and a verifier by adding a verification objective to the standard RL objective and achieves over 20% higher accuracy on MATH problems with parallel sampling and 8−32× more efficient test-time compute scaling compared to base RL methods. It also generalizes well to harder and out-of-domain tasks.

**Questions To Authors:**

1.I'm curious about how the method in the paper compares with PPO's value model in terms of performance and efficiency.
2.Section 5.2 seems to have little to do with the main theme of the paper. There are many existing works on the choice of TTS (test-time scaling) strategies. Can you explain the inevitable connection between this section and the idea of training the verifier and reasoner on the same model?

**Reasons To Accept:**

1.The experiments are very comprehensive and fully support the arguments of the paper.
2.This study holds certain significance for the LLM reasoning.

**Reasons To Reject:**

1.In Figure 1, Base RL uses LLM-as-a-judge and RL-V uses a  trained verifier. Is the performance and efficiency gap obtained in this way fair? Especially considering that this model is only 1.5 B, is it reasonable to use R1 or GPT4 as the LLM-as-a-judge?
2.The writing of the article is rather confusing. There is very little content about the experimental setup and training part. For example, the settings of each experiment in Section 5.1, such as "Easy-to-Hard Generalization", I couldn't find them in the article at all.
3.The idea is simple. The key lies in how to train the reasoner and verifier on the same model, but the article fails to highlight this point.

---

> ### Author Response · Authors · 2025-06-01
> **New Experiments and Clarifications**
>
> We thank the reviewer for their helpful comments and are confident they can be used to further improve the paper. We appreciate that the reviewer finds that our experiments are very comprehensive and fully support the arguments in the paper, while holding a certain significance for the LLM reasoning community.
>
> ## Using LLM-as-a-Judge for Base RL + New 7B results
>
> Our goal is to propose an alternative to the RL pipeline that co-trains for verification. Since we conduct a compute-matched analysis, the only way to obtain values from the base RL method is by LLM-as-a-Judge (prompting). We propose that this comparison is fair because we show that allocating some compute during RL fine-tuning to verification is beneficial for inference scaling at no additional cost. For example, since training is done in a wall-clock matched setting, RL^V sees less generations during fine-tuning than RL but still learns a capable reasoner policy. Having a separate verifier, even with LLM-as-a-Judge requires extra memory and compute. Note that we have added new results for a trained separate verifier in a reply to Reviewer 1 (https://openreview.net/forum?id=kVOrGZM5N7&noteId=0JrrRSGFd6). We also include majority voting as a strong value-free baseline.
>
> While we expect R1 or GPT4 to perform better than untrained 1.5B Qwen models when prompted for verification, this wouldn’t be a fair comparison since much more compute/data has been used to train these models. Co-training for verification in large-scale and frontier models like these should further improve verification and reasoning ability. To get an idea about the capability of frontier models as LLM Judges on math problems and the related challenges, we refer the reviewer to [Zheng et al. 2023]. In addition, [Singhi et al. 2025] show the value of fine-tuning large LLMs for verification.
>
> Further, to address your concern that larger LLMs will be more capable LLM-as-a-Judge verifiers, we have trained Qwen2.5-Math-7B using Leave-one-out PPO and Leave-one-out PPO^V.
>
> The results are summarized in the following figure: https://postimg.cc/bDHBr7r2
>
> Here, we see an improvement in LLM-as-a-Judge from the 1.5B -> 7B scale (50.4% -> 53.3% balanced verifier classification accuracy on MATH500 and 50.0% -> 54.4% on AIME’24) but co-training for verification still results in much higher verification accuracy.
>
> ## How to train a unified reasoner and verifier: Training and evaluation details
>
> Thank you for the feedback on the writing. We’d be happy to clarify any details about the experimental setup or training.
>
> All training is done on Hendrycks MATH. The "Easy-to-Hard Generalization" demonstrates that the RL^V reasoner and verifier generalize to more difficult problems by evaluating on MATH^2, a more difficult benchmark combining 2 math domain skills [Shah et al. 2025]. We use a generation length of 1024 for MATH and 2048 for all other benchmarks. We use a standard protocol to extract answers from \boxed{} and parse them using Math-Verify.
>
> With regards to the training setup, here are some more details that enable co-training. A linear ramp up was for both learning rate and verification coefficient, such that the maximum value is reached ~¾ of the way through training. We do a hyperparameter search for learning rate for all methods and use standard settings for other hyperparameters. To ensure both reasoner and verifier objectives are trained jointly, we include both gradients in a single batch using gradient accumulation. For a given reasoner batch, the verifier gradient is computed by class-balancing right/wrong answers and computing the loss, oversampling the class with less right/wrong answers. We ensure the batch size is large enough that the resampling is reasonable. This ensures 100% of the data generated during RL is used for verification. We use an asynchronous infrastructure with 1 generation GPU and 3 training GPUs. All training for RL^V and baselines is wall-clock matched.
>
> We believe this idea’s simplicity is a strength. It allows value-free RL to scale inference efficiently without adding compute or memory overhead. We will gladly edit the manuscript with these and any other details you think would help make the paper more complete.

---

> > ### Author Response · Authors · 2025-06-01
> > **New Experiments and Clarifications**
> >
> > ## New baseline: PPO as a verifier
> >
> > Thank you for the suggestion; this is a natural baseline. Recent work argues credit assignment from PPO’s value function is inaccurate and poorly reflects the probability of producing a correct answer [Kazemnejad et al. 2024]. This is a major contributor to the recent rise in popularity of “value-free” methods in RL fine-tuning [DeepSeek-AI 2025].
> >
> > We ran this experiment with Qwen2.5-Math-1.5B in a standard PPO setup. Since PPO’s value function provides token-level values and we need solution-level scores for re-ranking, we need to aggregate these values somehow. We find averaging the values over the solution as opposed to taking the value of the last token gives the best results.
> >
> > Results on MATH500: https://postimg.cc/mzKYdYxS
> >
> > This result suggests that PPO’s value function can act as a verifier (note the small gap between majority voting and weighted voting) but is less effective than RL^V.
> >
> > ## Section 5.2 in relation to the rest of the work
> >
> > We agree the main purpose of our work is not to demonstrate which TTS strategies are optimal. The objectives of this section were to
> > 1. explain how our RL^V verifiers can be used
> > 2. mention that we try multiple TTS options for fair and comprehensive evaluation across all methods. Without this, comparisons between methods could be biased toward specific inference strategies
> > 3. familiarize new readers with the tradeoffs of different TTS strategies
> >
> > If the reviewer feels there is any way to reformulate the section to better meet these ends or has any other suggestions, we are happy to incorporate them.
> >
> > ## References
> >
> > Lianmin Zheng, Wei-Lin Chiang, Ying Sheng, Siyuan Zhuang, Zhanghao Wu, Yonghao Zhuang, Zi Lin, Zhuohan Li, Dacheng Li, Eric P. Xing, Hao Zhang, Joseph E. Gonzalez, and Ion Stoica. Judging LLM-as-a-Judge with MT-Bench and Chatbot Arena. arXiv preprint arXiv:2306.05685, 2023.
> >
> > Nishad Singhi, Hritik Bansal, Arian Hosseini, Aditya Grover, Kai-Wei Chang, Marcus Rohrbach, and Anna Rohrbach. When To Solve, When To Verify: Compute-Optimal Problem Solving and Generative Verification for LLM Reasoning. arXiv preprint arXiv:2504.01005, 2025.
> >
> > Vedant Shah, Dingli Yu, Kaifeng Lyu, Simon Park, Jiatong Yu, Yinghui He, Nan Rosemary Ke, Michael Mozer, Yoshua Bengio, Sanjeev Arora, and Anirudh Goyal. AI-Assisted Generation of Difficult Math Questions. arXiv preprint arXiv:2407.21009, 2025.
> >
> > Amirhossein Kazemnejad, Milad Aghajohari, Eva Portelance, Alessandro Sordoni, Siva Reddy, Aaron Courville, and Nicolas Le Roux. VinePPO: Unlocking RL Potential For LLM Reasoning Through Refined Credit Assignment. arXiv preprint arXiv:2410.01679, 2024.
> >
> > DeepSeek-AI. DeepSeek-R1: Incentivizing Reasoning Capability in LLMs via Reinforcement Learning. arXiv preprint arXiv:2501.12948, 2025.

---

> > > ### Author Response · Authors · 2025-06-05
> > > **Follow-Up**
> > >
> > > Thank you for your valuable review. We have tried to address your concerns by adding new experiments and clarifications. Do you have any remaining questions or comments? We want to ensure there is enough time to address them.

---

> > ### Author Response · Authors · 2025-06-08
> > **Follow-Up**
> >
> > Thank you again for your review. To address your concerns, we have added 2 new experiments:
> > 1. results for the 7B model
> > 2. trained and evaluated PPO as a verifier per your request
> >
> > We have also added comprehensive details about the training and evaluation setup and clarified the role of section 5.2 in our work as requested. Given that there are only 3 days remaining in the discussion period, please let us know if you have remaining questions or if there are any concerns that were not addressed.

---

> > > ### Comment · Reviewer_f9F3 · 2025-06-09
> > >
> > > Thanks for the author's reply and sorry for my delayed response. The author's reply addressed my concerns. I am willing to raise my score.

---

### Official Review · Reviewer_BvLu · 2025-05-12

**Rating:** 6
**Confidence:** 3
**Ethics Flag:** 1

**Summary:**

This paper proposes to add an additional task to RL training of LLMs. In addition to training to produce answers with maximum reward, train the same model to predict a binary success at the end of the sequence, in the form of an appended prompt asking if the solution is correct and a yes/no answer.
The paper shows two interesting aspects of this simple addition: it actually helps the model produce more accurate answers first shot and the confidence of "yes" of the model can be use as a score for the generated answer and help select the best-of-n answer in a pool of solutions.

**Questions To Authors:**

Figure 7 tends to show that leave-one-out-PPO works better for your approach, why not use that one everywhere else instead of showing  GRPO results in figure 2,5,and 6?

You have produced both long CoT results and BoN selection that both improve accuracy at test-time but did not compare or discussed the best way to improve the accuracy for a given compute budget. Section 5.2 touches on that point but does not answer many important questions that practitioners might have:
- If latency is not a problem, and we only care about compute (number of generated tokens), is it always better to scale the CoT length or can it be beneficial to use your selection method and select a BoN answer? If BoN is viable, at what N value and for what CoT length?
- If latency is a problem, since BoN can be parallelized, it might be a viable solution even with a higher token budget. In that case where could a trade-of between CoT and BoN be?

In figure 5 it would be interesting to include an oracle pass@n curve to have an idea of the performance ceiling of the proposed BoN selection mechanism.

Typo line 51 "time-time" -> "test-time"

**Reasons To Accept:**

This work proposes a simple approach with limited drawbacks (not much complexity added to the implementation, not much additional compute involved) and clear benefits.

Experiments showing comparisons with long and short CoT and different RL methods are interesting and cover many of the questions one could have about the proposed approach.

Results shown here spark interest for further research.

**Reasons To Reject:**

It is not always clear in the paper how the experiments and results shown in the paper are selected. For example figure 4 shows only Leave-one-out-PPO results while figure 2,5,6,  only GRPOv results.
AIME 24 is used in most graphs without error bars. This is a benchmark of only 30 questions which can lead to high variance of the results and limited confidence in the conclusions.

---

> ### Author Response · Authors · 2025-06-03
> **Rebuttal**
>
> We thank the reviewer for their thoughtful evaluation of our work. We appreciate that they recommend acceptance and find our work thorough, with limited drawbacks and clear benefits while sparking interest for future research.
>
> ## Oracle Pass@N in Figure 5
>
> Thank you for the suggestion. We include the plot here: https://postimg.cc/PPHjc0QB
>
> Note that for GPQA, the coverage is high since answer options are provided as part of the prompt.
>
> ## When does parallel inference scaling become more FLOP efficient than CoT scaling?
>
> This is a very important question. To the best of our knowledge, this kind of study is generally lacking in the field. [Singhi et al. 2025] (released Apr 1, 2025) contains some discussion about this question and a concurrent paper [Qu et al. 2025] (released Mar 10, 2025) looks at the majority vote accuracy on math tasks after budget forcing the model to different token lengths.
>
> At the lower token budget allowed by our compute infrastructure, it is better to scale CoT length. For example, generating N solutions with a P parameter model and sequence length L costs roughly 2PLN FLOPs [Hoffmann et al. 2022]. Consider Figure 6 at 2^5 solutions per problem. Doubling the generation length 1024 -> 2048 increases the accuracy by ~12% while sampling 2^6 solutions increases the accuracy by ~2%. This tradeoff appears artificially starker since a generation length of 1024 tokens is too low to capture the full benefits of parallel scaling. As generation length increases, this tradeoff should increasingly favor parallel scaling (note the slopes of the curves in Figure 6 increase with generation length meaning parallel scaling becomes more efficient). Also, models may overthink at longer generation lengths, resulting in diminishing returns on accuracy when scaling CoT length [Chen et al. 2025][Su et al. 2025].
>
> Additionally, unlike FLOPs, latency per token increases with generation length [Dao et al. 2022] and, as you mention, parallel scaling can be done with lower latency when there is access to a lot of compute, making it more valuable in a large scale setting.
>
> ## Trade-off between CoT and parallel scaling assuming parallelization
>
> Good question. This is essentially asking: if we can freely parallelize our inference scaling, what would the trade off be in terms of CoT length? The answer is in our results for long CoT (Figure 2, 6). CoT scaling is orthogonal to our method and any amount of parallel sampling will improve the results.
>
> ## Why GRPO in Figures 2, 4 and 6
>
> We focus on GRPO for the long CoT R1-Distill model. The ablations on base RL method are conducted with the Qwen2.5-Math model, where we find Leave-one-out PPO is best. There is potential to further improve our long CoT results by using Leave-one-out PPO. We have started these experiments. They may not finish by the end of the discussion period but we will certainly include them in the camera ready version.
>
> ## Error Bars for AIME
>
> To compute our inference scaling curves for AIME, we use a methodology from [Zhang et al. 2025]. We generate 2048 solutions per problem and compute all metrics for 200 subsets at each parallel inference budget before averaging them. This results in very accurate accuracy estimates for inference scaling. We conducted repetitions to verify this and found consistent results.
>
> Thank you for the deep-dive into our work and let us know if you have any more questions or suggestions.

---

> > ### Author Response · Authors · 2025-06-03
> > **Rebuttal**
> >
> > ## References
> >
> > Xingyu Chen, Jiahao Xu, Tian Liang, Zhiwei He, Jianhui Pang, Dian Yu, Linfeng Song, Qiuzhi Liu, Mengfei Zhou, Zhuosheng Zhang, Rui Wang, Zhaopeng Tu, Haitao Mi, and Dong Yu. Do NOT Think That Much for 2+3=? On the Overthinking of o1-Like LLMs. arXiv preprint arXiv:2412.21187, 2025.
> >
> > Jinyan Su, Jennifer Healey, Preslav Nakov, and Claire Cardie. Between Underthinking and Overthinking: An Empirical Study of Reasoning Length and Correctness in LLMs. arXiv preprint arXiv:2505.00127, 2025.
> >
> > Jordan Hoffmann, Sebastian Borgeaud, Arthur Mensch, Elena Buchatskaya, Trevor Cai, Eliza Rutherford, Diego de Las Casas, Lisa Anne Hendricks, Johannes Welbl, Aidan Clark, et al. Training Compute-Optimal Large Language Models. arXiv preprint arXiv:2203.15556, 2022.
> >
> > Nishad Singhi, Hritik Bansal, Arian Hosseini, Aditya Grover, Kai-Wei Chang, Marcus Rohrbach, and Anna Rohrbach. When To Solve, When To Verify: Compute-Optimal Problem Solving and Generative Verification for LLM Reasoning. arXiv preprint arXiv:2504.01005, 2025.
> >
> > Tri Dao, Daniel Y. Fu, Stefano Ermon, Atri Rudra, and Christopher Ré. FlashAttention: Fast and Memory-Efficient Exact Attention with IO-Awareness. arXiv preprint arXiv:2205.14135, 2022.
> >
> > Lunjun Zhang, Arian Hosseini, Hritik Bansal, Mehran Kazemi, Aviral Kumar, and Rishabh Agarwal. Generative Verifiers: Reward Modeling as Next-Token Prediction. arXiv preprint arXiv:2408.15240, 2025.
> >
> > Yuxiao Qu, Matthew Y. R. Yang, Amrith Setlur, Lewis Tunstall, Edward Emanuel Beeching, Ruslan Salakhutdinov, and Aviral Kumar. Optimizing Test-Time Compute via Meta Reinforcement Fine-Tuning. arXiv preprint arXiv:2503.07572, 2025.

---

> > ### Author Response · Authors · 2025-06-05
> > **Follow-Up**
> >
> > We appreciate your effort in writing the review and believe it has helped us strength our work. Given the discussion period is ending soon (including a weekend), we want to ensure there is enough time to address further questions or comments you may have. Please let us know if there are any remaining concerns.

---

> > ### Comment · Reviewer_BvLu · 2025-06-07
> >
> > I believe the oracle pass@n curve deserves to be discussed more. It shows a lot of potential for improvement in answer selection methods.
> >
> > The discussion brought by the author about scaling flops is interesting but does not bring a definitive answer.
> >
> > The trade-off between CoT and parallel that I suggested does not suppose free compute. I am considering compute on one axis, latency on the other and want to see in what region of the compute/latency space it is preferable to use BoN vs CoT for a given level of performance.
> >
> > Thank you for clarifying your approach to AIME, it sounds like a good process that should allow you to compute and report standard error of the resulting accuracy.
> >
> > Some of my questions were answered. I still think this paper is above acceptance threshold but I have not been convinced to raise the score higher.

---

> > > ### Author Response · Authors · 2025-06-11
> > >
> > > > I believe the oracle pass@n curve deserves to be discussed more. It shows a lot of potential for improvement in answer selection methods.
> > >
> > > We would be happy to include this in the appendix and add some discussion. The oracle pass@n corresponds to the attainable accuracy by a perfect verifier that always picks the correct answer if it exists. Our verifier is a 1.5B model trained only on data logged during the RL run. A reasonable comparison for our method is a separate verifier trained on the same data during the RL run. These results (see https://openreview.net/forum?id=kVOrGZM5N7&noteId=0JrrRSGFd6) show a small gap in verifier performance between unified and separate verifiers indicating not much is lost by unified co-training.
> > >
> > > > The discussion brought by the author about scaling flops is interesting but does not bring a definitive answer.
> > >
> > > We are glad you found this discussion interesting. While we agree the FLOP efficiency comparison of parallel vs. sequential compute is an important question, we believe it is a question with a large scope that deserves its own study. This kind of study would be applicable to any type of parallel scaling with any method of training a verifier, whereas our main contribution is to demonstrate how to co-train a unified verifier during RL. We've only found a partial exploration of this topic in Wang et al. (2025), published on May 28, 2025.
> > >
> > > While we would be glad to explore this comparison further, doing so rigorously would require scaling to significantly longer generation lengths. Unfortunately, this is currently infeasible given our hardware constraints: training memory scales quadratically with sequence length.
> > >
> > > That said, we emphasize that parallel and sequential compute are **complementary**. Scaling both together will push the frontier of performance under a fixed FLOP budget. For instance, in regimes where CoT scaling plateaus (or declines due to overthinking), parallel scaling is expected to become more FLOP efficient. Parallel compute also offers clear latency advantages.
> > >
> > > > Thank you for clarifying your approach to AIME, it sounds like a good process that should allow you to compute and report standard error of the resulting accuracy.
> > >
> > > We appreciate the suggestion. We have estimated the standard error on AIME accuracy to be approximately ±0.2%.
> > >
> > > Please let us know if you have any further questions or concerns.
> > >
> > > **References**
> > >
> > > Zili Wang, Tianyu Zhang, Lei Zhu, Haoli Bai, Lu Hou, Shiming Xiang, Xianzhi Yu, & Wulong Liu. (2025). Faster and Better LLMs via Latency-Aware Test-Time Scaling.

---

### Official Review · Reviewer_SGCT · 2025-05-13

**Rating:** 7
**Confidence:** 3
**Ethics Flag:** 1

**Summary:**

This paper proposes a method for training LLM-based verifiers used for test-time scaling. The proposed method trains a single LLM as both a reasoner and a verifier by training its reasoning ability using a value-free reinforcement learning method like GRPO and also training its verification ability using labeled data generated in the reinforcement learning procedure. The proposed approach is cost-efficient since we can use the abandoned data generated in a reinforcement learning procedure. Moreover, it is resource-efficient since we can use a single LLM as both a reasoner and a verifier. Experimental results on multiple tasks show that using a learned LLM in tasks with test-time scaling techniques can improve the performance in multiple task settings, compared with a simple majority voting and a method using the LLM-as-a-Judge method.

**Questions To Authors:**

I want to know the details of the training procedure for optimizing the unified objective. Specifically:
1. Does the proposed method use all the data generated by an LLM reasoner during the RL process?
2. According to the explanation in line 112, training generative verifiers needs a class-balanced dataset. How does the proposed method prepare it?

**Reasons To Accept:**

1. The paper is clearly written and easy to read.
2. The idea of using abandoned data generated in value-free reinforcement learning is interesting. I like the simplicity of the proposed approach.
3. The paper evaluates the proposed approach in various settings to prove its effectiveness.

**Reasons To Reject:**

The figures of experimental results are appropriately placed and easy to understand, but they do not seem exhaustive since the results of some combinations of RL methods and test datasets are not reported. For example, in-distribution results with the MATH2 dataset are not reported in the paper. I think the paper should report the results for other combinations of RL/task/LLM in the appendix.

---

> ### Author Response · Authors · 2025-06-03
> **Rebuttal**
>
> We thank the reviewer for their thoughtful review and appreciate that they recommend our work be accepted and find it simple, effective, and well-written with exhaustive evaluations.
>
> ## Complete results
>
> Thank you for the suggestion, we will include such a table in appendix. There may be a bit of confusion since MATH^2 is only a test set so there is no training data to do in-distribution tests for it.
>
> ## Details on training
>
> These are great questions:
> 1. We use all of the data generated during RL. We take all solutions generated to form a verification dataset.
> 2. To ensure both reasoner and verifier objectives are trained jointly, we include both gradients in a single batch using gradient accumulation. For a given reasoner batch, the verifier gradient is computed by class-balancing right/wrong answers and computing the loss, oversampling the class with less right/wrong answers. We ensure the batch size is large enough that the resampling is reasonable.
>
> Thank you again and let us know if you have any more questions or suggestions for improving our work.

---

> > ### Author Response · Authors · 2025-06-05
> > **Follow-Up**
> >
> > Thank you for recommending our paper for acceptance. Do you have any remaining questions or comments?

---

> > > ### Comment · Reviewer_SGCT · 2025-06-06
> > >
> > > Thank you for addressing my concerns. I have no further questions.

---

### Official Review · Reviewer_Sqzo · 2025-05-21

**Rating:** 6
**Confidence:** 3
**Ethics Flag:** 2

**Summary:**

The propose jointly training LLMs for both reasoning with RL and verification with an SFT based generative verifier. The compare their approach to vanilla GRPO, PPO leave-one-out, and vine PPO on MATH 500, MATH^2, GPQA Diamond, and AIME 2024. They find that their joint training approach improves performance substantially when using parallel sampling. For a fixed sequential test-time budget, it also outperforms the baselines at pass@1, suggesting that the verification objective makes for useful representation learning. They finally conduct ablations with the loss weight for balancing the reasoning and verifier training.

**Questions To Authors:**

See the reasons to reject section.

**Reasons To Accept:**

* The paper is mostly well written and the problem is important: finding ways to improve reasoning models by combining sequential and parallel test-time compute is an impactful research direction.
* They carry out fairly thorough evaluations on multiple benchmarks and do a reasonable job of ablating relevant design decisions.
* I thought the finding in 5.2 about BoN performing worse on short CoT and better on long CoT is very interesting!

**Reasons To Reject:**

* Based on their setup in the intro/abstract I assumed they were doing some kind of chain of thought verifier e.g. “Unlike traditional value functions predicting only scalar rewards, generative verifiers leverage the LLM’s generation capabilities.” Actually, three approach also just generates a scalar reward, the only difference is that it uses the LLM's yes no token heads instead of a special token head to achieve this. I found this kind of misleading. They highlight the chain of thought case as future work, but I think this could be clearer in the intro what they are actually doing. The main contribution is arguably less so generative RM and more so, jointly training the model for both capabilities.
* We can still do majority voting without a learned verifier, so I would contest the claim that existing reasoning models cannot do parallel test-time compute at all. That being said, majority voting is somewhat less general because it requires being able to do an exact match or equivalence between final answers, which is not always the case. They could potentially spell this out more in the paper.
* I'm confused about how they do the weighted best-of-N process with the baseline algorithms? Are you using an external verifier or is it the same one from the RL^V model?
* One baseline I would be curious about is: how well does the parallel test-time compute perform if you sample from the base model (e.g. the model you started with before finetuning) and then use the RL^V verifier to re-rank? This seems like an important attribute to understand: how much of the performance is coming from the verifier quality verses the generation quality?
* A lot of the experiments feel not too insightful, or its not clear what to takeaway, e.g. why is leave one out PPO so different from GRPO^V in Section 5.4?

---

> ### Author Response · Authors · 2025-06-01
> **New Experiments and Clarifications**
>
> We thank the reviewer for their thoughtful comments. We are glad they find this research direction impactful and that they appreciate our efforts to conduct thorough evaluations and ablations.
>
> ## New ablation: Base model with RL^V Verifier
>
> Thanks for the suggestion. We ran this ablation with the Qwen2.5-Math-1.5B reasoner and Leave-one-out PPO^V verifier.
>
> The results are in the figure: https://postimg.cc/KRLS73CF
>
> This demonstrates that during RL^V, there is a drastic improvement in the quality of the reasoner itself (Pass@1: 34.1% -> 58.6% in right figure) in addition to verification ability (weighted vote for Qwen2.5-Math-1.5B + RL^V is ~5% higher than majority vote in left figure). It also shows that Pass@1 is not the most important factor when scaling inference, since a capable verifier is able to bring Qwen2.5-Math-1.5B close to Leave-one-out PPO^V at high inference compute budgets (76% vs. 79% at 512 solutions) despite a ~25% lower Pass@1.
>
> This ablation also shows the generalization capabilities of our verifier since it has been trained on on-policy samples from the RL trained version of the model (not the solution distribution from the base model), yet it still works well.
>
> ## How to do value-based test-time scaling with value-free RL
>
> We use LLM-as-a-Judge to get values for best-of-N and weighted voting for the baseline. This is because we propose an alternative to the RL fine-tuning pipeline that enables verification. The only way the base value-free RL pipeline allows for verification without modifications in a compute-matched setting is by prompting the model to obtain scores for a given solution (LLM-as-a-Judge). We use the prompt in Appendix D.3 and use the probability of “Yes” as the verification score. By contrast, training a separate verifier would require additional training compute and memory during inference.
>
> However, we see the value of training a separate verifier to contextualize our results. We have trained a separate verifier using Qwen2.5-Math-1.5B on the data logged from the base leave-one-out PPO run and report the results below. Note that this requires double the memory and additional training compute compared to all other methods. We also rerun the head ablations with leave-one-out PPO for consistency.
>
> | Verifier Method                         | Verifier MATH500 Balanced Classification Accuracy (%) | Reasoner MATH500 Accuracy (Pass@1) |
> |----------------------------------------|--------------------------------------------------------|-------------------------------------|
> | Separate Verifier                      | 82.2                                                   | -                                   |
> | Unified RL^V Verifier                  | 80.1                                                   | 58.6                                |
> | RL + Verification Head (Regression)   | 54.1                                                   | 47.7                                |
> | RL + Verification Head (BCE)          | 55.3                                                   | 51.6                                |
> | RL (LLM-as-a-Judge)                   | 50.4                                                   | 55.6                                |
>
> This demonstrates that not much is lost in terms of verifier quality by doing our version of unified training which requires half the memory and less training compute.
>
> ## Clarification: CoT Verification
>
> The main contribution of our work is joint co-training of the unified reasoner and verifier during RL. We don’t do CoT verification but we still train a generative verifier. It is a Yes/No verifier which corresponds to the base GenRM setting in [Zhang et. al 2024]. Sorry for the confusion, we will make this clearer in the introduction and the abstract.
>
> Unified co-training with CoT verification poses challenges that we leave for future work. Instead, we focus on thoroughly investigating how to jointly train a reasoner and verifier within RL. It might be interesting for the reviewer that a future paper by [Liu et al 2025] released on May 19, 2025 (1.5 months after COLM deadline), tackles some of these challenges by training a unified CoT verifier during RL and provides more evidence that co-training for verification during RL is a fruitful direction

---

> > ### Author Response · Authors · 2025-06-01
> > **New Experiments and Clarifications**
> >
> > ## Clarification: Value-free parallel test-time compute with majority voting
> >
> > We agree existing reasoning models can scale parallel test-time compute using majority voting. We did not intend to claim that they cannot, and indeed have this as a baseline already in all of our experiments.
> >
> > Instead, we make the claim that scaling parallel test-time compute without a verifier is suboptimal and less efficient. This finding is supported by our own experiments (see Figures 4 and 5) and prior work. It is the main finding of recent work by [Setlur et al. 2024] which provides strong theoretical evidence for this claim but this is also observed in many empirical studies [Lightman et al. 2023] [Cobbe et al. 2021] [Luo et al. 2024].
> >
> > We would be happy to make this more clear in the paper.
> >
> > ## Insight from experiments and why GRPO differs from Leave-one-out PPO
> >
> > We appreciate the reviewer’s feedback. Some core insights are as follows:
> > 1. Our choice of generative verifier is necessary for unified training since verification heads degrade the policy and result in a bad verifier (Figure 7a)
> > 2. Verification training has the potential to improve generation quality, highlighting the potential use of verification for representation learning
> > 3. This kind of training generalizes robustly across distribution shifts (MATH -> MATH2, GPQA), base RL algorithm (GRPO, Leave-one-out PPO, VinePPO), long and short CoT models and scale (see new 7B experiments)
> > 4. Leave-one-out PPO pairs best with RL^V, balancing reasoning and verification objectives better than alternatives like GRPO
> >
> > This point about Leave-one-out PPO and GRPO is a great question and was a point of discussion amongst the authors. We hypothesize it’s related to how the KL is treated in both loss functions. Many modern implementations of Leave-one-out PPO (eg. RLOOTrainer in TRL) incorporate the KL as part of the reward while GRPO uses an additional term summed in the loss. It’s possible this makes the algorithm more flexible for adding verification training. We are continuing to investigate this question.
> >
> > Thank you again for your comments and please let us know if your concerns were addressed or if there are any more changes we can make to improve the paper.
> >
> > ## References
> >
> > Hunter Lightman, Vineet Kosaraju, Yuri Burda, Harrison Edwards, Bowen Baker, Teddy
> > Lee, Jan Leike, John Schulman, Ilya Sutskever, and Karl Cobbe. Let’s verify step by step.
> > In The Twelfth International Conference on Learning Representations, 2023.
> >
> > Karl Cobbe, Vineet Kosaraju, Mohammad Bavarian, Mark Chen, Heewoo Jun, Łukasz Kaiser, Matthias Plappert, Jerry Tworek, Jacob Hilton, Reiichiro Nakano, Christopher Hesse, and John Schulman. Training Verifiers to Solve Math Word Problems. arXiv preprint arXiv:2110.14168, 2021.
> >
> > Xiaoyuan Liu, Tian Liang, Zhiwei He, Jiahao Xu, Wenxuan Wang, Pinjia He, Zhaopeng Tu, Haitao Mi, and Dong Yu. Trust, But Verify: A Self-Verification Approach to Reinforcement Learning with Verifiable Rewards. arXiv preprint arXiv:2505.13445, 2025.
> >
> > Lunjun Zhang, Arian Hosseini, Hritik Bansal, Mehran Kazemi, Aviral Kumar, and Rishabh Agarwal. Generative Verifiers: Reward Modeling as Next-Token Prediction. arXiv preprint arXiv:2408.15240, 2025.
> >
> > Liangchen Luo, Yinxiao Liu, Rosanne Liu, Samrat Phatale, Meiqi Guo, Harsh Lara, Yunxuan Li, Lei Shu, Yun Zhu, Lei Meng, Jiao Sun, and Abhinav Rastogi. Improve Mathematical Reasoning in Language Models by Automated Process Supervision. arXiv preprint arXiv:2406.06592, 2024.

---

> > > ### Author Response · Authors · 2025-06-05
> > > **Follow-Up**
> > >
> > > Thank you for taking the time to review our work and providing insightful feedback. Do you have any remaining concerns or more advice on improving our work? Given that rebuttal period is ending soon, we would like to ensure there is enough time to address any further questions or comments.

---

> > > > ### Comment · Reviewer_Sqzo · 2025-06-07
> > > >
> > > > I appreciate the additional experiments to address some of my concerns. And I appreciate the references to related work to contextualize your approach. I think addressing the items mentioned will make the paper better. I would highlight the fact that while separating the two can result in better results, it adds a large additional cost. I am willing to raise my score a point as long as these items are addressed in the camera ready.

---

> > > > > ### Author Response · Authors · 2025-06-07
> > > > >
> > > > > Thank you for the constructive feedback. It has helped improve the paper. We would be happy to include these in the camera ready.

---

### Author Response · Authors · 2025-06-11
**AC Comment**

After useful and fruitful discussions with all reviewers, we seem to have satisfied all concerns and **all reviewers recommend acceptance**:

Our work has been well-received for its impactful direction and thoroughness, with reviewers highlighting the paper as "clearly written" and appreciative of our evaluation of the "approach in various settings to prove its effectiveness”. They recognized it for sparking "interest for further research”. Furthermore, our experiments have been described as "very comprehensive and fully support the arguments of the paper" holding "significance for LLM reasoning."

We thank all reviewers for their time and effort and note the biggest changes from the discussion period.

**Added experiments, ablations and baselines**

We have added the following experiments, ablations and baselines during the rebuttal period:
- A PPO baseline showing that PPO’s value function can act as a verifier, though **it is less effective than RL^V**
- An ablation using the RL^V verifier with the base model, showing that both generation are verification objectives are optimized during training and that our verifier generalizes to the solution distribution of the base model
- A baseline using a separately trained verifier on the same data, showing **unified co-training gives comparable results at a much lower cost**
- An experiment validating our method on the Qwen2.5-Math-7B model

---

### Decision · Program_Chairs · 2025-07-08

**Decision:**

Accept

**Comment:**

This paper proposes to jointly train a single LLM to serve both as a generator and a (generative) verifier for LLM reasoning tasks. While prior work (e.g., GRPO, Leave-one-out PPO)has shown that value functions in PPO can introduce bias and has advocated for value-free reinforcement learning, this paper demonstrates that a jointly trained verifier can enable efficient and generalizable test-time scaling.
The reviewers largely agree that the paper is mostly well-written (Sqzo, SGCT), tackles an important problem (Sqzo, f9F3), and is supported by extensive experiments (Sqzo, SGCT, BvLu, f9F3).

Several reviewers suggested additional experiments to strengthen the empirical validation: evaluating a base model with an RL^V verifier (Sqzo), reporting other combinations of RL/task/LLM (SGCT), reporting Oracle Pass@N (BvLu), and using PPO’s value model as a verifier (f9F3). The authors have included these experiments during the rebuttal phase, which helps address many of these concerns.

To sum up, I would recommend acceptance for this paper.